# A Comprehensive Transcriptional Signature in Pancreatic Ductal Adenocarcinoma Reveals New Insights into the Immune and Desmoplastic Microenvironments

**DOI:** 10.3390/cancers15112887

**Published:** 2023-05-24

**Authors:** Irene Pérez-Díez, Zoraida Andreu, Marta R. Hidalgo, Carla Perpiñá-Clérigues, Lucía Fantín, Antonio Fernandez-Serra, María de la Iglesia-Vaya, José A. Lopez-Guerrero, Francisco García-García

**Affiliations:** 1Bioinformatics and Biostatistics Unit, Principe Felipe Research Center (CIPF), 46012 Valencia, Spain; 2Biomedical Imaging Unit FISABIO-CIPF, Fundación para el Fomento de la Investigación Sanitaria y Biomédica de la Comunidad Valenciana, 46012 Valencia, Spain; 3IVO-CIPF Joint Research Unit of Cancer, Príncipe Felipe Research Center (CIPF), 46012 Valencia, Spain; zandreu@fivo.org (Z.A.);; 4Department of Physiology, School of Medicine and Dentistry, University of Valencia, 46010 Valencia, Spain; 5Laboratory of Molecular Biology, Fundación Instituto Valenciano de Oncología, 46009 Valencia, Spain; 6Department of Pathology, Medical School, Catholic University of Valencia, 46001 Valencia, Spain

**Keywords:** pancreatic ductal adenocarcinoma, desmoplasia, immune system, heterogeneity cancer, biomarkers, molecular profile, meta-analysis, transcriptomics, prognosis, meta-analysis

## Abstract

**Simple Summary:**

Pancreatic ductal adenocarcinoma (PDAC) is a highly lethal disease with a few curative options. Desmoplastic stroma and immune system evasion in PDAC represent challenges to the success of therapeutic strategies are used to suitably treat other tumor types. Characterizing the PDAC microenvironment (including the immune environment) remains critical to developing safe and efficient therapies. Here, we present a comprehensive meta-analysis identifying 1153 significantly dysregulated genes, which mainly impact extracellular matrix remodeling and the immune system. We identify two signatures of twenty-eight immune-related genes and eleven stroma-related genes influencing PDAC patients’ survival. Additionally, five immune genes are associated with PDAC prognosis for the first time.

**Abstract:**

Pancreatic ductal adenocarcinoma (PDAC) prognoses and treatment responses remain devastatingly poor due partly to the highly heterogeneous, aggressive, and immunosuppressive nature of this tumor type. The intricate relationship between the stroma, inflammation, and immunity remains vaguely understood in the PDAC microenvironment. Here, we performed a meta-analysis of stroma-, and immune-related gene expression in the PDAC microenvironment to improve disease prognosis and therapeutic development. We selected 21 PDAC studies from the Gene Expression Omnibus and ArrayExpress databases, including 922 samples (320 controls and 602 cases). Differential gene enrichment analysis identified 1153 significant dysregulated genes in PDAC patients that contribute to a desmoplastic stroma and an immunosuppressive environment (the hallmarks of PDAC tumors). The results highlighted two gene signatures related to the immune and stromal environments that cluster PDAC patients into high- and low-risk groups, impacting patients’ stratification and therapeutic decision making. Moreover, *HCP5*, *SLFN13*, *IRF9*, *IFIT2*, and *IFI35* immune genes are related to the prognosis of PDAC patients for the first time.

## 1. Introduction

Pancreatic ductal adenocarcinoma (PDAC) is the most common type of pancreatic cancer, representing over 80% of all diagnosed pancreatic neoplasms. This highly lethal cancer has a poor prognosis, with a median survival rate of fewer than six months, although its five-year survival rate increased to 12% in recent years [1]. While it is currently the third leading cause of cancer-related deaths worldwide [1], the yearly increase in its incidence may make PDAC the second leading cause of cancer-related deaths by 2030 [1]. The absence of reliable biomarkers for effective screening and early diagnosis at the pre-symptomatic stages when treatments function most effectively represents a primary reason why most PDAC cases remain incurable. Currently, most patients present locally advanced (30–35%) or metastatic (50–55%) PDAC at diagnosis [2].

In advanced-stage PDAC patients, curative surgery remains impossible, and systemic therapeutic options (including immunotherapy) remain limited and ineffective [3]. Among the solid tumors, PDAC represents an immunologically “cold” tumor characterized by sparse T cell infiltration [4,5]; in contrast, immunologically “hot” tumors (such as melanoma) suffer from a high neoantigen load and immune cell infiltration [6]. PDAC tumors possess distinctive features such as an extracellular matrix (ECM) composition and a fibrotic stroma, which make it highly desmoplastic and significantly influence immune responses [7]. PDAC cells strongly interact with the surrounding microenvironment, which includes components such as immune cells, cytokines, metabolites, fibroblasts, and hyaluronan. These interactions create a highly fibrotic and active organized stroma (desmoplastic stroma) and an immunosuppressive environment that makes PDAC invasive and highly resistant to immunotherapy [5,8]; therefore, the characterization of the stroma and tumor immune microenvironment in PDAC patients represents a critical step in developing more effective therapeutic strategies. In the last few years, several investigations have focused on studying gene expression in PDAC to better understand the molecular composition of this devastating cancer and identify different molecular subtypes of pancreatic cancer that improve the stratification of patients for clinical strategies [9,10]. Bailey and colleagues defined four molecular subtypes of pancreatic cancer: squamous, pancreatic progenitor, immunogenic and aberrantly differentiated endocrine exocrine (ADEX) [10], while Moffitt’s group identified two stromal subtypes that were defined as “normal” and “activated” [9]. Nevertheless, in clinical practice, it is difficult to perform this broad molecular test on each patient. Therefore, and despite these new insights in pancreatic cancer, the diagnostic and prognostic outcomes of PDAC patients are extremely poor compared to those of other types of cancers. Additionally, new studies need to be conducted to understand the extreme complexity of PDAC and find simpler genetic signatures that can be incorporated into clinical practice and improve the clinical setting for PDAC patients and families.

We aimed to understand the stroma and tumor immune microenvironments of PDAC patients by retrieving and analyzing transcriptomic data from 21 different studies (representing a population of 922 samples; 320 controls and 602 cases) from the Gene Expression Omnibus (GEO)-NCBI and ArrayExpress data repositories. Through meta-analysis, we identified a series of gene signatures with survival prognostic value that may play a significant role in therapeutic decision making for PDAC patients, including five genes not previously related to PDAC survival. We also provide a friendly user web tool with detailed and interactive visualization of our comprehensive meta-analysis results.

## 2. Materials and Methods

For all bioinformatics and statistical analyses, we employed R software v. 4.1.3 [11] (Appendix A details the R packages and versions).

### 2.1. Study Search and Selection

Publicly available datasets were collected from GEO-NCBI [12] and ArrayExpress databases [13]. Data available in the Cancer Genome Atlas (TCGA) [14] were excluded from the original search with the purpose of using this dataset as an external cohort for survival analysis. Following the Preferred Reporting Items for Systematic Reviews and Meta-Analyses (PRISMA) guidelines [15], a systematic search of published studies was conducted in 2021 (period: 2002–2021). The protocol has not been registered. Three researchers in the study conducted the literature search (C.P.C., L.F., and I.P.D.), and the consistency of the review and selection procedures used was evaluated and confirmed. A broad search was performed using the MeSH (Medical Subject Headings) thesaurus keyword “pancreatic cancer”, after which stringent filters were applied. The final inclusion criteria were:Normal and PDAC samples available.RNA extracted directly from human pancreas biopsies.Patients had not undergone treatment before biopsy.Sample size > 4 for PDAC and control groups.

Finally, normalized gene expression from twenty-seven microarray studies (GSE86436, GSE71989, GSE62452, GSE62165, GSE60979, GSE56560, GSE55643, GSE46234, GSE43795, GSE43288, GSE41368, GSE32676, GSE28735, GSE27890, GSE22780, GSE19650, GSE18670, GSE16515, GSE15471, GSE1542, GSE11838, GSE102238, GSE101448, E-MTAB-3365, E-MTAB-1791, E-MEXP-950, and E-EMBL-6) and the count matrices of two RNA-sequencing (RNA-seq) (GSE119794 and GSE136569) datasets were retrieved for further analysis.

### 2.2. Individual Preprocessing and Analysis

Datasets were individually analyzed in two steps: preprocessing and differential expression analysis.

The nomenclature of clinical variables included in each study was standardized for data preprocessing, and then, exploratory analysis was performed. Prior to exploratory analysis, RNA-seq raw count matrices were normalized using the trimmed mean of m values from the edgeR package [16,17]. The normalization method performed by the original authors for each microarray dataset was assessed, and the matrices were log2 transformed when necessary. Exploratory analysis included expression boxplots, unsupervised clustering, and principal component analysis (PCA) to detect patterns of expression between samples and genes and the presence of batch effects in each study.

Differential gene expression analyses were performed in R using limma (v. 3.48.3) [18], and a paired sample design was implemented in those datasets where applicable. Differentially expressed genes were identified using *p* values with Benjamini-Hochberg correction [19] for a false discovery rate (FDR) at a significance level of 0.05.

### 2.3. Gene Expression Meta-Analysis

Gene expression analysis results were integrated into a meta-analysis using the DerSimonian & Laird random effects model [20], considering individual study heterogeneity. This model considers the variability in individual studies by increasing the weights of studies with less variability when meta-analysis results are computed.

A total of 24,365 genes were evaluated. *p* values, FDR-corrected *p* values, the logarithm of Fold Change (log2FC), and 95% confidence intervals of log2FC were calculated for each evaluated gene, and both funnel and forest plots were computed for each gene. These representations were assessed for possible biased results, where log2FC represents the effect size of a function, and the standard error of the log2FC serves as a study precision measure [21]. Genes were considered significant when FDR < 0.05, absolute log2FC > 0.6, and were measured in at least eleven studies. Sensitivity analysis (leave-one-out cross-validation [22]) was conducted for each significant gene to verify alterations in the results, owing to the inclusion of any study.

Statistically significant results from the gene expression meta-analysis were functionally enriched by over-representation analysis (ORA) using clusterProfiler [23,24] and ReactomePA [25]. Gene Ontology (GO) terms [26,27] and Reactome pathway [28] enrichment were performed following this approach. Only those functions and pathways with more than ten differentially expressed genes found in the gene set were considered. Functional enrichment was explored and visualized with the *rrvgo* package [29].

### 2.4. Web Tool

To make the data and results of our research widely accessible, a web tool was developed using the shiny package in R. The tool was developed in a user-friendly manner, allowing users to navigate and interact with the data. Users can then select different variables and parameters to visualize the data in numerous ways. The tool also includes interactive plots and tables to display the analysis results. The web tool is hosted on a secure server and is regularly maintained to ensure stability and performance. The source code for the tool is also publicly available and can be accessed through our GitHub repository: https://github.com/ipediez/ShinyReport (accessed on 1 May 2023).

### 2.5. Survival Analysis

RNA-seq expression data and metadata from patients in the Pancreatic adenocarcinoma (PAAD) TCGA cohort were downloaded from cBioPortal [30]. Z-scores of RNA-seq expression were used for survival analysis. For each analyzed gene, samples were divided into two groups based on their expression levels. Samples with expression Z-scores below the lower quartile were classified as having low levels of expression, whereas samples exceeding the upper quartile were classified as having high levels of expression. Forty-five samples with high levels of expression and forty-five samples with low levels of expression were included for survival analysis. Gene-wise Kaplan–Meier survival analysis compared the low-level and high-level expression groups. This method estimates the probability of survival over time based on the expression levels of the gene of interest. The log-rank test was used to compare the survival curves between distinct groups of samples.

For risk-score-based survival, genes were tagged as highly expressed for a given sample when the expression levels were above the upper quartile. Then, samples were clustered into “high-risk” and “low-risk” groups based on the number of highly expressed genes. The cutoff was set as the median of the highly expressed genes in each sample. Furthermore, a proportional hazard model using Cox regression was implemented to study the impact of clinicopathological variables on survival and evaluate the contribution of the risk score in a multivariate model.

## 3. Results

We performed a systematic review and differential gene expression analysis of PDAC transcriptomic studies from the GEO-NCBI [12] and ArrayExpress [13] databases to explore the stroma and immune environments in PDAC patients. We then integrated the results of each differential gene expression analysis into a meta-analysis. The biological context of the meta-analysis results was explored via functional enrichment using an ORA of GO terms and pathways (Figure 1). Finally, we conducted survival analysis to explore the impact of specific candidate genes on patients’ outcomes.

### 3.1. Systematic Review

The systematic review identified 143 non-duplicated studies. Then, we excluded studies with samples from patients under cancer treatment and studies where the sample size was less than four in the PDAC or the control group, resulting in a subset of twenty-nine studies (Figure 2). We discarded eight studies after exploratory analysis, giving a final set of twenty-one homogeneous and comparable studies for further analysis. The selected studies included 922 samples (320 controls and 602 cases). Although most studies did not include relevant sample metadata, we assessed the clinical characteristics when they were available. Appendix A contain further information regarding the selected studies and clinicopathological characteristics of the study population.

### 3.2. Integration of Differential Expression Profiles

Exploratory analysis found abnormal normalization or a lack of annotation in eight studies, which we excluded from further analysis (listed in Appendix A). Then, we performed the independent differential gene expression analysis of each study and meta-analysis for 24,365 genes evaluated in the different datasets, including every gene found in at least two studies. We considered results with an FDR < 0.05, an absolute log2FC > 0.6, and those evaluated in at least eleven studies to be significant; overall, 1153 genes met these criteria (Figure 3; further details are given in Appendix A).

We noted the presence of genes encoding ECM components (e.g., collagens, fibronectin, laminin, and stratifin), proteoglycans (e.g., versican), cell adhesion molecules, integrins, matrix metallopeptidases, and additional peptidases and enzymes that impact mechano-contractility, epithelial tension, and the stiffness of the tumoral stroma, which can promote tumor progression and resistance to therapy (Figure 4). Table 1 displays the twenty genes with the highest and lowest log2FC values from the meta-analysis; these genes mainly play roles in ECM remodeling, desmoplasia, metabolism, and the immune system. Appendix A reports a complete list of significantly affected genes.

We performed ORA using GO biological process terms to identify the possible implications of 1153 significantly differentially expressed genes in the PDAC samples. We considered only those biological processes with at least ten associated genes and an adjusted *p* value under 0.05. We found 546 over-represented biological processes among the over-expressed genes and 40 biological processes over-represented among the under-expressed genes (Appendix A). ORA revealed the enrichment of terms related to the tumor microenvironment (Figure 5), with GO terms related to the immune system, cell adhesion, and ECM remodeling/degradation. Of note, additional over-represented functions were related to metastasis (vascularization, cell migration, collagen, mesenchymal transition, cell proliferation, and peptidyl modifications) [7,31].

### 3.3. Interactive Tool for Results Visualization

The web tool contains comprehensive information regarding the data and results of the meta-analysis of gene expression. The application includes tables and plots for the differential expression results of twenty-one datasets included in the study and meta-analysis results. Statistical indicators, such as the log odds ratio, confidence intervals, and adjusted *p* values, are provided to estimate each study’s global expression and specific contribution. The web tool is available online: https://bioinfo.cipf.es/MetaPDAC/ (accessed on 1 January 2020).

### 3.4. Immune System: A Functional Overview in PDAC

To focus our analysis on the tumor immune microenvironment, we extracted a consensus list of genes related to the immune system and inflammation from NCBI and GO databases (mainly framed in the categories of HLA, interleukin, CD, interferon, chemokine, and S100 genes, Appendix A). Considering an FDR threshold of 0.05 and an absolute fold change greater than 0.6, we discovered the significant differential expression of 322 immune genes in our meta-analysis results. To explore the functional involvement of these results, we performed ORA on this group of genes using GO biological process terms and Reactome pathways. We considered significant functional terms with at least ten associated genes and an adjusted *p* value < 0.05. We discovered the over-representation of thirty-three GO terms and twenty-seven pathways among the over-expressed immune-related genes and none when under-expressed genes were analyzed. The enriched terms suggest the increased activity of neutrophil-related immune response, the negative regulation of cell killing, interferon signaling, and an antigen presentation via major histocompatibility complex II.

### 3.5. Immune and Stromal Survival Signatures Impact PDAC Prognosis

We explored the 322 differentially expressed immune-related genes and identified a set of 70 genes of particular interest in our experimental research (Table 2). We performed survival analysis using the TCGA PAAD cohort for each of these genes and found statistically significant differences in twenty-eight genes (*IFI27*, *IL1R2*, *IL1RN*, *IL1RAP*, *IL18*, *IL22RA1*, *HCP5*, *SLFN13*, *CD58*, *CD109*, *IFI44L*, *IFI16*, *IFITM1*, *IFIT1*, *IFIT3*, *IRF9*, *IFIT2*, *IFI35*, *CXCL10*, *CXCL5*, *CXCL9*, *S100P*, *S100A6*, *S100A2*, *S100A16*, *S100A11*, *S100A14*, and *S100A10*), which shared a pattern: a higher expression in patients associated with a lower rate of survival. As far as we are aware, this is the first time that *HCP5*, *SLFN13*, *IRF9*, *IFIT2*, and *IFI35* have been related to prognosis value in PDAC patients (Appendix A).

We analyzed genes that displayed statistical significance as a “signature,” dividing the samples into high-risk and low-risk groups based on the number of highly expressed genes (above the upper quartile). We set the median (six highly expressed genes) as the cutoff value to divide the samples into groups. Interestingly, patients in the high-risk group possessed shorter survival times than those in the low-risk group did (*p* value < 0.0001, Figure 6A). Furthermore, we studied the effect of this signature in a multivariate Cox model including age, alcoholic history, the presence of chronic pancreatitis, diabetes diagnostic, tumor grade, and the AJCC classification of a metastatic tumor and a residual tumor as covariates. The proposed signature was the only variable with *p* value < 0.05 and showed a hazard ratio of 2.36 (Appendix A). We then analyzed the co-occurrence of highly expressed genes in the samples, finding two main co-occurrence groups that related to high-risk patients: i) the interferon gene family (IFN genes) and ii) the S100 and IL genes (*S100A14*, *S100A16*, *S100A6*, *S100A11*, *IL1R2*, *IL1RN*, and *S100P*) (Figure 6B).

To explore how a desmoplastic environment can affect patients’ survival, we employed an homologous approach using genes related to ECM remodeling (Table 1). We discovered eleven genes whose survival analysis showed statistically significant differences (*CEACAM5*, *CEACAM6*, *FN1*, *GJB2*, *GPRC5A*, *LAMB3*, *LAMC2*, *SFN*, *SLC6A14*, *TSPAN1*, and *VCAN*). Again, we divided the samples into high-risk and low-risk groups using the median of the number of highly expressed genes as the cutoff value (median = 3). Patients with high levels of expression in three or more genes from the signature presented lower survival times than those with fewer highly expressed genes did (*p* value = 0.00012, Figure 7A). Of note, we distinguished a cluster of co-occurrence of patients with high levels of *GJB2*, *FN1*, and *VCAN* at the same time (Figure 7B).

Finally, we performed comparative analysis between the immune and stromal survival signatures identified in our work and other signatures generated in previous works for patient stratification [9,10]. These results provided insight into the level of intersection between this group of signatures (Appendix A).

## 4. Discussion

Using comprehensive meta-analysis, we explored the immune environment and desmoplastic stroma of PDAC tumors to contribute to a deeper understanding of tumorigenesis and the design of effective therapeutic strategies, such as immunotherapies. ECM components from the desmoplastic stroma tightly interact with the immune environment and contribute to immune evasion by modulating immune cell infiltration, thus influencing cell proliferation, tumor progression, and overall survival [32,33]. The meta-analysis and ORA results characterized differences in the gene-expression landscape of PDAC tumors and identified more than 1000 dysregulated genes, most of them with immune system- and desmoplasia-related roles. We discovered thirty-nine genes (twenty-eight immune-related genes and eleven stroma-related genes) that impact PDAC patients’ survival.

Among the top forty dysregulated genes (Table 1), we observed the upregulation of collagens (*COL11A1* and *COL10A1*), which influence immune infiltration and chemoresistance and confer a poor prognosis [34,35,36]. PDAC patients also presented with upregulated periostin expression, which has been linked to a shorter overall survival [37], and *cystatin SN*, which contributes to pancreatic cancer cell proliferation and may represent a potential biomarker for the early detection of pancreatic cancer [38]. Stratifin and matrix metallopeptidase 1 also appeared to be upregulated in PDAC patients; stratifin stimulates matrix metallopeptidase 1 expression in fibroblasts, contributing to remodel ECM [39]. The increased expression of fibronectin in the PDAC stroma has also been reported. The observed upregulation of *cathepsin E* and *sulfatase 1* expression in the PDAC microenvironment might also benefit the development of therapeutic strategies with polymer drug conjugates since they may contribute to drug release [40,41,42].

The analysis of the top forty dysregulated genes also provided evidence for the downregulation of genes coding for proteolytic enzymes released by the pancreas (e.g., *chymotrypsin*, *chymotrypsinogen*, *lipases*, and *phospholipases*). Pancreatic cancer cells express around 20% of chymotrypsin C normal cells expression, with this enzyme participating in cancer cell apoptosis and migration [43]. A recent report suggested that a combination of trypsinogen and chymotrypsinogen displayed an anti-tumorigenic potential [44].

Focusing on the immune environment, PDAC tumors develop a wide range of mechanisms to evade the immune system (e.g., a low level of expression of HLA antigens, immunosuppressive signals that inhibit natural killer and T cell functions, and the presence of immunosuppressive cells). This creates an immunotolerant environment in which the immune system of PDAC patients does not robustly recognize and target cancer cells [45]. We explored the expression of seventy genes of particular interest, including those from the HLA, interleukin, CD, interferon, chemokine, and S100 categories. The survival analysis of these genes in the TCGA PAAD cohort identified a twenty-eight immune-related gene signature with a prognostic value that was used to cluster PDAC patients into high-risk and low-risk groups.

The proposed signature possessed significance in univariate and multivariate Cox models with clinicopathological variables, significantly adding statistical power to the survival analysis. This signature could aid in the stratification of patients (Figure 8) who could benefit from immunotherapeutic strategies, given that it could contribute to distinguishing “cold” PDAC tumors (characterized by the low presence of T cells (CD8+) and natural killer cells, high presence of immunosuppressive cell populations, and poor prognoses and responses to immunotherapy) from “hot tumors” (with an opposite profile) [46,47]. We uncovered two high gene-expression co-occurrence patterns, one composed of IFN genes and the other of S100/IL genes. The IFN signaling pathways participate in PDAC development, while the over-expression of S100 genes blocks the infiltration and cytotoxic activity of CD8+ T cells, and the low level of expression of IL1RN and IL1R2 has been associated with increase survival in PDAC patients [48,49,50].

To the best of our knowledge, this is the first report of data suggesting a link between the *HCP5*, *SLFN13*, *IRF9*, *IFIT2*, and *IFI35* immune genes and PDAC prognosis, presenting the discriminatory power of clustering PDAC patients. The remaining genes of the immune gene signature have been individually associated with PDAC or other cancers, with data suggesting that their overexpression could impact patients’ diagnosis, prognosis, and response to treatment [51,52,53,54,55,56]; however, we report that a joint gene expression signature of these genes impacts PDAC patients’ survival.

Focusing on the PDAC stroma, the altered genes include several types of collagens, fibronectins, and proteolytic enzymes, such as metalloproteases and peptidases (Table 1 and Appendix A), which significantly contribute to ECM composition and stromal remodeling and support desmoplasia and immunosuppression [57]. The survival analysis of significantly dysregulated stromal gene expression from the meta-analysis of the TCGA PAAD cohort revealed a gene signature with prognostic capacity that clustered PDAC patients into high-risk and low-risk groups. We observed a co-occurrence pattern in high-risk patients, indicating a subgroup of PDAC patients with a high level of expression of *GJB2*, *FN1*, and *VCAN* genes. These results indicate stromal heterogeneity in PDAC [58] and the need to characterize it to stratify patients (Figure 8).

With respect to other dysregulated genes, the upregulation of *CEACAM5* and *CEACAM6* represents an early event in pancreatic carcinogenesis, with these genes being candidates for immunotherapies [59,60,61]. Furthermore, laminins *LAMBC2* and *LAMB3* support cancer progression and resistance to gemcitabine—one of the main chemotherapeutics used in PDAC patients [62,63]. In general, the association of the stroma signature with a poor prognosis is consistent with the one described in previous studies for each gene: *CEACAM5* [64], *CEACAM6* [65], *FN1* [66], *GJB2* [67], *GPRC5A* [68], *LAMB3* [69,70], *LAMC2* [69,70], *SFN* [71], *SLC6A14* [72], *TSPAN1* [73], and *VCAN* [66].

With respect to other similar approaches, we are aware of two additional studies in which expression datasets were integrated to explore the nature of the PDAC in depth: one by Gooneskere and colleagues, who integrated six PDAC and three other pancreatic carcinomas datasets [74], and one by Irigoyen and colleagues, who integrated two peripheral blood datasets [75]. Both approaches integrate different datasets at the gene level to increase the number of samples and perform unique DGE analysis. In contrast, our approach analyzed each dataset independently, and then integrated the results, evaluating their robustness. From the experimental design point of view, both studies differ greatly from ours, since Grooneskere et al.’s one is not specifically focused on PDAC, and Irigoyen et al.’s one does not analyze pancreatic tissue. From a methodological point of view, our study contributes to a more profound and robust analysis of the PDAC expression landscape by integrating data after DGE analysis had been performed, thus avoiding the necessity to control heterogeneity among studies and retaining the full potential of biological differences.

Other molecular studies based on whole transcriptome and genomic analyses of pancreatic tumors have found specific gene signatures that identify different molecular subtypes [9,10]. However, the aim of this study was not to identify molecular subtypes, such as in the cited works. The immune signature or the stromal signature presented in this work establishes patient survival groups (high-risk group and low-risk group), which could help practitioners to decide if the patient could benefit from immunotherapy, for example, or not. Intersection analysis indicated that there is hardly any overlap between the gene signatures found in our study and the signatures described by Bailey et al. [10] or Moffitt et al. [9], as shown in the supplemental analysis (Appendix A). Therefore, the proposed gene signatures show subtype-independent survival value and display a reasonable number of genes for them to be translated to clinics. Nevertheless, more and deeper studies are needed for this purpose. Additionally, the works by Moffitt et al. and Bailey et al. are enormously rich and provide comprehensive molecular stratification to facilitate personalized treatment and the identification of therapeutic targets. Unfortunately, extensive molecular analyzes are difficult to translate to clinical practice for individual patients.

A potential limitation of our study has been the relative heterogeneity among the sample sizes and sequencing platforms used. The meta-analysis methodology, which integrates data groups and provides results with higher statistical power and precision [76,77], addresses this issue by independently comparing each study and combining the results. A lack of clinical and/or molecular information in most studies, such as survival time, stage condition, or molecular pattern, represents an additional limitation. We employed TCGA data for survival analysis, but additional analyses should integrate other covariates of interest in the study.

Finally, we provided an interactive web tool that allows users to explore our results, facilitating the accessibility, transparency, and reusability of our research. Overall, the web tool provides a detailed and interactive visualization of the meta-analysis results, allowing users to further explore and understand the gene expression patterns identified in the studies. Other functionalities include the capability to customize and filter the data to further investigate specific aspects of the analysis in more detail. In this manner, we aim to align our research with the FAIR principles to share our data in a way that can be of further use to the scientific community who studies this aggressive and lethal tumor.

## 5. Conclusions

Therapeutic strategies to overcome the immune microenvironment and the desmoplastic stroma barriers remain limited and generally unsuccessful. This study performs a comprehensive transcriptional meta-analysis of the molecular PDAC environment. The results highlight the relevance of the interaction between the immune system and stroma, revealing an impact on patients’ survival. The identified gene signatures provide new insights into the potential therapeutic targets for this deadly disease that can help to stratify its heterogeneity. Future studies are needed to explore the benefits of targeting the immune and stromal microenvironments as a treatment strategy for PDAC.

## Figures and Tables

**Figure 1 cancers-15-02887-f001:**
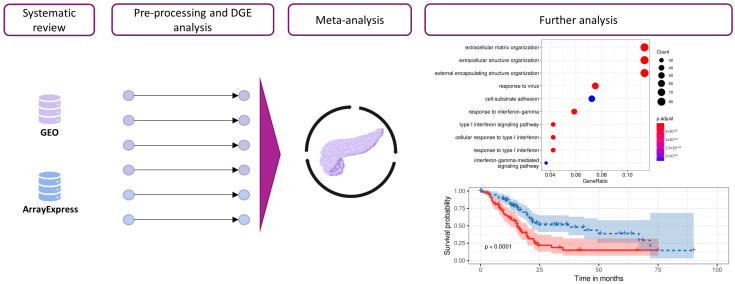
Workflow and analysis design. Relevant studies from the GEO-NCBI and ArrayExpress databases were retrieved, and data exploration and preprocessing were then performed. After DGE analysis, the results from different studies were integrated into a gene meta-analysis. Functional profiling methodologies were applied to explore the biological implications of the results.

**Figure 2 cancers-15-02887-f002:**
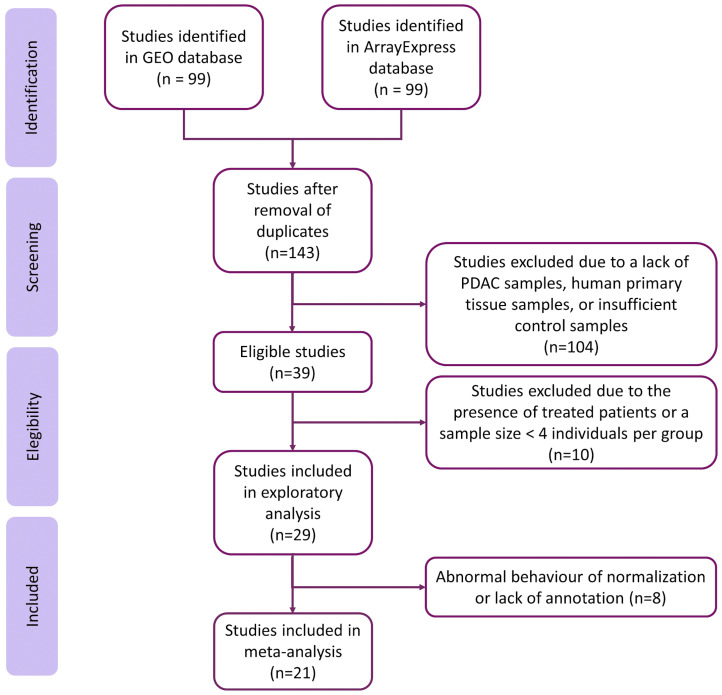
Flow of information through the distinct phases of the systematic review, following PRISMA Statement guidelines.

**Figure 3 cancers-15-02887-f003:**
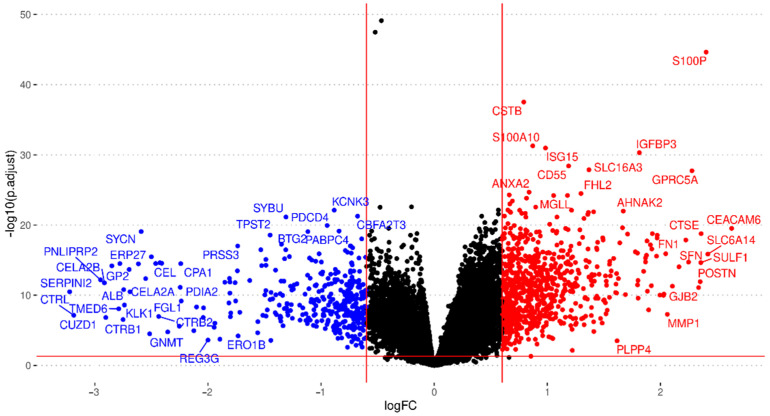
Volcano plot summarizing the gene expression meta-analysis. Significantly over-expressed genes are shown in red, and significantly under-expressed genes are shown in blue (FDR < 0.05; absolute log2FC > 0.6). Genes that do not show significant differential expression are represented in black. Only genes found in at least eleven studies are shown.

**Figure 4 cancers-15-02887-f004:**
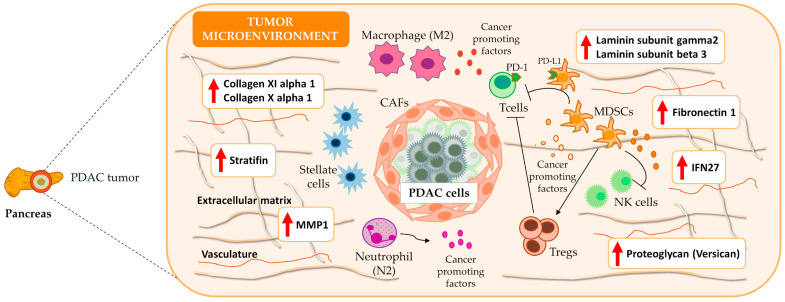
Overview of PDAC microenvironment. Meta-analysis results indicated an overexpression of several ECM components, e.g., stratifin, fibronectin 1, different laminin subtypes (gamma2 and beta3), collagens, and proteoglycans that characterize the dense and desmoplastic stroma of PDAC tumors. Additionally, the results highlight the presence of immune components such as IFN27, which contribute to an increase in the number of M2 macrophages and a decrease in the number of CD8+ T cells. Therefore, the desmoplastic stroma and the immune system favor immune tolerance and poor prognosis in PDAC. The red upward-pointing arrows denote genes exhibiting significant overexpression in the conducted meta-analysis. IFN27: interferon alpha inducible protein; MMP1: matrix metallopeptidase 1; NK cells: natural killer cells; T cells: T effector lymphocytes; Tregs: T regulatory lymphocytes T.

**Figure 5 cancers-15-02887-f005:**
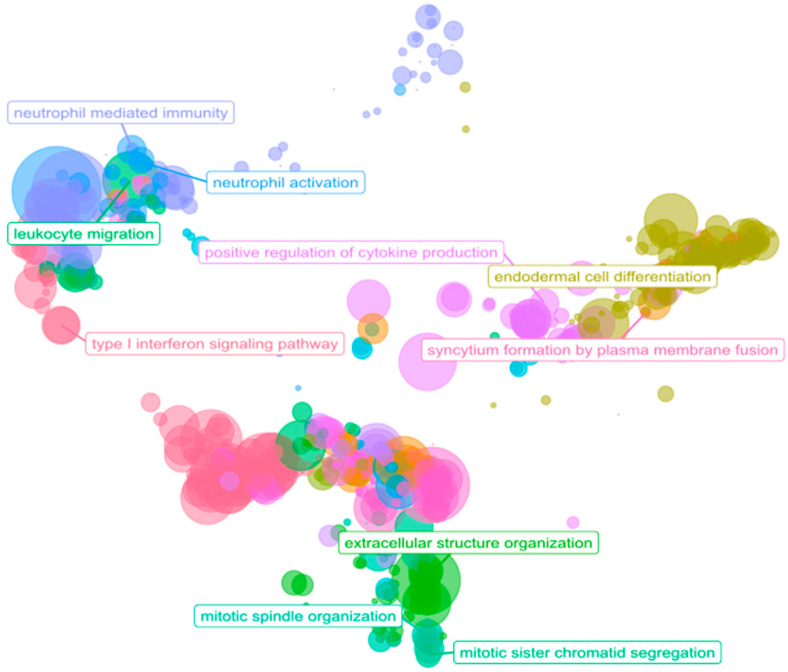
Scatter plot of ORA results. The scatterplot reports the GO biological process representative terms after redundancy reduction in a two-dimensional space derived from the semantic similarities between GO terms. The dot size represents the number of biological processes related to a GO term. The parent terms of the main clusters are labeled.

**Figure 6 cancers-15-02887-f006:**
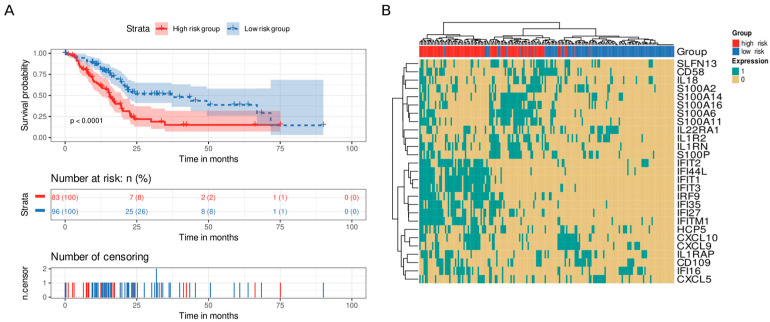
Survival analysis of immune system genes. A twenty-eight gene signature clustered patients into high-risk or low-risk groups based on the number of highly expressed signature genes in their transcriptomic profile. Patients with at least six highly expressed genes were classified as having a high risk, whereas those with five or fewer were classified as having a low risk. (**A**) Kaplan–Meier curve. Patients from the high-risk group (red) had shorter survival times than patients from the low-risk group did (blue). Below, the number of still alive patients and percentage in each group at 0, 25, 50, 75, and 100 months, and the censored events. (**B**) Heatmap demonstrating the patterns of high expression between genes and samples. Gene expression was coded as 1 for a sample above the upper quartile.

**Figure 7 cancers-15-02887-f007:**
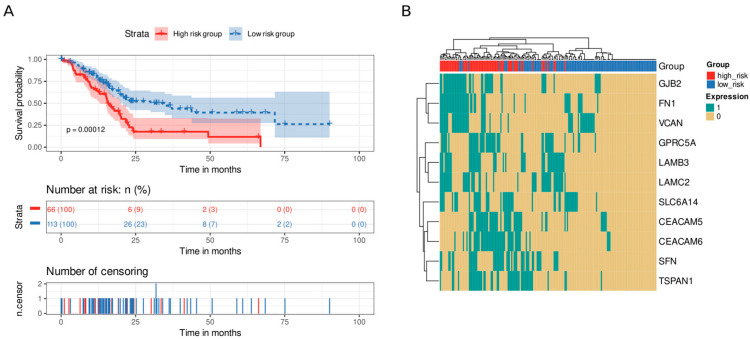
Survival analysis of ECM remodeling genes. An eleven-gene signature clustered patients into high-risk or low-risk groups based on the number of highly expressed signature genes in their transcriptomic profile. Patients with at least three highly expressed genes were classified as having a high risk, whereas those with five or fewer were classified as having a low risk. (**A**) Kaplan–Meier curve. Patients from the high-risk group (red) had shorter survival times than patients from the low-risk group did (blue). Below, the number of still alive patients and percentage in each group at 0, 25, 50, 75, and 100 months, and the censored events. (**B**) Heatmap demonstrating the patterns of high expression between genes and samples. Gene expression was coded as 1 for a sample above the upper quartile.

**Figure 8 cancers-15-02887-f008:**
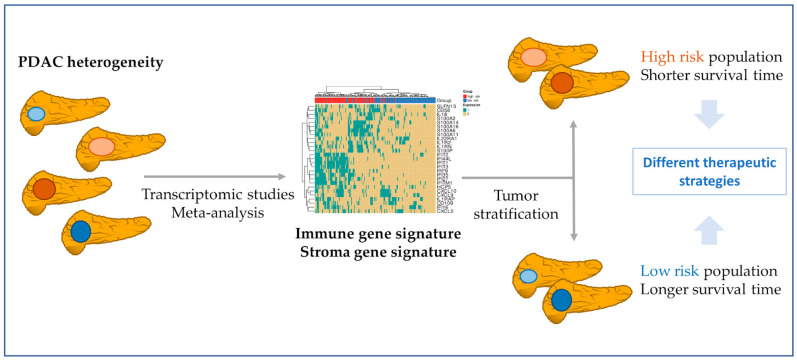
Patient stratification based on PDAC molecular features. Meta-analysis from transcriptomic studies allows a better understanding of the PDAC environment. In this study, the found gene signatures might contribute to the stratification of PDAC patients. In a first step, the immune or the stroma gene signatures can divide patients into high- and low-risk populations. After, with a focus on the immune signature co-occurrence, patients could be divided into those with more S100/IL genes and those with a more IFN expressed genes. The knowledge about these molecular features of PDAC tumors may guide the design of more effective therapeutic strategies.

**Table 1 cancers-15-02887-t001:** Top twenty genes up- and down-regulated in PDAC patients.

Gene Symbol	Gene Name	Expression Level	Function
*CEACAM6*	CEA cell adhesion molecule 6	UP	EMR
*SLC6A14*	Solute carrier family 6 member 14	UP	EMR
*S100P*	S100 calcium-binding protein P	UP	EMR
*CTSE*	Cathepsin E	UP	EMR
*SULF1*	Sulfatase 1	UP	EMR
*POSTN*	Periostin	UP	EMR
*GJB2*	Gap junction protein beta 2	UP	EMR
*GPRC5A*	G protein-coupled receptor class C group 5 member A	UP	EMR
*SFN*	Stratifin	UP	EMR
*FN1*	Fibronectin 1	UP	EMR
*LAMC2*	Laminin subunit gamma 2	UP	EMR
*CEACAM5*	CEA cell adhesion molecule 5	UP	EMR
*MMP1*	Matrix metallopeptidase 1	UP	EMR
*COL11A1*	Collagen type XI alpha 1 chain	UP	EMR
*TSPAN1*	Tetraspanin 1	UP	EMR
*IFI27*	Interferon alpha inducible Protein 27	UP	IS
*CST1*	Cystatin SN	UP	EMT
*LAMB3*	Laminin subunit beta 3	UP	EMR
*COL10A1*	Collagen type X alpha 1 chain	UP	EMR
*VCAN*	Versican	UP	EMR
*CTRB2*	Chymotrypsinogen B2	DOWN	EMR
*PLA2G1B*	Phospholipase A2 group IB	DOWN	Metabolism
*CTRC*	Chymotrypsin C	DOWN	EMR
*GNMT*	Glycine N-methyltransferase	DOWN	Metabolism
*AQP8*	Aquaporin 8	DOWN	H_2_O_2_ transport
*SYCN*	Syncolin	DOWN	Exocytosis
*CPA2*	Carboxypeptidase A2	DOWN	Metabolism
*CELA2A*	Chymotrypsin-like elastase 2A	DOWN	EMR
*GP2*	Glycoprotein 2	DOWN	Metabolism
*KLK1*	Kallikrein 1	DOWN	Serine protease
*ALB*	Albumin	DOWN	Oncotic pressure
*CTRB1*	Chymotrypsinogen B1	DOWN	EMR
*ERP27*	Endoplasmic reticulum protein 27	DOWN	Lipid and protein synthesis
*TMED6*	Transmembrane p24 trafficking protein 6	DOWN	Insulin secretion
*PNLIPRP1*	Pancreatic lipase-related protein 1	DOWN	Metabolism
*CUZD1*	CUB and zona pellucida-like domain 1	DOWN	EMR and IS
*CELA2B*	Chymotrypsin-like elastase 2B	DOWN	EMR
*PNLIPRP2*	Pancreatic lipase-related protein 2	DOWN	Metabolism
*CTRL*	Chymotrypsin-like	DOWN	EMR
SERPINI2	Serpin family I member 2	DOWN	Protease inhibitor

EMR = ECM remodeling; IS = immune system; EMT = epithelial–mesenchymal transition.

**Table 2 cancers-15-02887-t002:** Subset of immune-related genes.

Functional Group	Genes
HLA	*HLA-F*, *HLA-DRB5*, *HLA-B*, *HLA-A*, ***HCP5***, *HLA-DRA*, *HLA-DPA1*, *HLA-DQB1*, *HLA-DQA1*, *HLA-DMB*, *HLA-DRB1*, *HLA-G*, *HLA-DPB1*, *SLFN12*, ***SLFN13***, and *SLFN11*
Interleukin	***IL1R2***, ***IL1RN***, ***IL1RAP***, *IL7R*, *IL2RG*, *IRAK3*, ***IL18***, *LIF*, and ***IL22RA1***
CD	***CD58***, ***CD109***, *CD52*, *CD53*, *CD74*, *CD14*, *CCDC80*, *CCDC141*, *CCDC69*, *DCDC2*, and *PDCD4*
Interferon	***IFI27***, ***IFI44L***, *IFI6*, *STING1*, ***IFI16***, ***IFITM1***, *ISG20*, ***IFIT1***, ***IFIT3***, *IFITM2*, ***IRF9***, ***IFIT2***, *IFNGR2*, *IFITM3*, and ***IFI35***
Chemokine	*CCL20*, *CCL18*, ***CXCL10***, ***CXCL5***, *CXCL8*, *CXCR4*, *CKLF*, ***CXCL9***, *CXCL3*, *CXCL14*, and *CXCL12*
S100	***S100P***, ***S100A6***, ***S100A2***, ***S100A16***, ***S100A11***, *S100A4*, ***S100A14***, and *S100A10*

Genes in bold possess statistically significant differences according to survival analysis.

## Data Availability

Publicly available datasets were analyzed in this study. Data are openly available in GEO and ArrayExpress, with the following accession numbers: GSE71989, GSE62452, GSE62165, GSE60979, GSE56560, GSE55643, GSE43795, GSE41368, GSE32676, GSE28735, GSE22780, GSE18670, GSE16515, GSE15471, GSE1542, GSE11838, GSE101448, GSE119794, GSE136569, E-MEXP-950, and E-EMBL-6.

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
