# Peer review of "A Comprehensive Transcriptional Signature in Pancreatic Ductal Adenocarcinoma Reveals New Insights into the Immune and Desmoplastic Microenvironments"

_cancers, 2023, doi:10.3390/cancers15112887_

Round 1

Reviewer 1 Report

In the metastudy by Perez-Diez et al, 21 datasets out of an initial 198 datasets are used for an analysis to stratify PDAC patients. In these datasets, over 1153 siginificantly regulated genes can be identified that have an impact on the tumor microenvironment (TME).

The TME is critical for drug resistance and the failure of immunotherapeutic approaches. Therefore, it is necessary to distinguish in which PDAC patients certain therapies could be used. Remodulation of immunosuppressive TME is a goal of several studies and resources to stratify PDAC patients according to TME are limited. Therefore, the approach shown here is relatively interesting but needs significant modifications and adaptations.

My suggestions on this: 

The introduction is not up to date. Current literature should be cited. For example, 5 year survival has been above 10% rather than 5% for some time. 

Also, PDAC subtypes and stroma subtypes that are already established should be introduced (e.g. Moffitt et al.). The approach is good in principle but lacks integrative analysis and comparison with established candidate subtype genes.

Both the immunogenic subtype (Bailey et al, etc.) and various stromal subtypes have already been identified (Moffitt et al, etc.). Is there overlap from already established candidate gene sets? 

If not, why is a difference apparent and what does this mean for (successful) stratification? Overlaps from QM, squamous and basal-like PDAC subtypes could already be shown, is this accordingly (stroma-associated subtypes) also possible here?

I am not qualified to assess the quality of English in this paper.

Reviewer 2 Report

A comprehensive transcriptional signature in pancreatic ductal adenocarcinoma reveals new insights into the immune and desmoplastic microenvironment Irene Pérez-Díez 1,2,3 †, Zoraida Andreu 3 †, Marta R. Hidalgo 1,3, Carla Perpiñá-Clérigues 1,3,4, Lucía Fantín 1 , Antonio Fernandez-Serra 3,5, María de la Iglesia-Vaya 2,3, José A Lopez-Guerrero 3,5,6 * and Francisco García-García 1,3 *

This manuscript performed a comprehensive in silico study that defined two transcriptional signatures related to immune and stroma milieu of PDAC. In addition, it created a web tool for easy access and review DEGs and associated survival and GO terms. It has values to the PDAC study community and thus worthy of the publication. Few minor suggestions as below.  

Minor

1.       five-year survival rate for pancreatic cancer now is 12% and is the third leading cancer death. Cancer statistics, Siegel, et al. 2023.

2.       logFC>0.6: to be more accurate, this is log2[FC]?

3.       Subtitle is replicated in section 2 methods and section 3 results, which caused confusion for reading. Please use sentences for subtitles in result section for easier reading.

4.       Please remove replicated description in methods and results.

Round 2

Reviewer 1 Report

The authors were able to clarify all concerns and addressed all points. Therefore, I believe the manuscript will be of interest to the readership of Cancers.